# A Rapid and Inexpensive PCR Test for Mastitis Diagnosis Based on NGS Data

**DOI:** 10.3390/pathogens13050423

**Published:** 2024-05-17

**Authors:** Agnieszka Kajdanek, Magdalena Kluska, Rafał Matusiak, Joanna Kazimierczak, Jarosław Dastych

**Affiliations:** Proteon Pharmaceuticals, Tylna 3a, 90-364 Łódź, Poland; magdalenakluska15@gmail.com (M.K.); rmatusiak@proteonpharma.com (R.M.); jkazimierczak@proteonpharma.com (J.K.); jdastych@proteonpharma.com (J.D.)

**Keywords:** bovine mastitis, mastitis diagnostics, subclinical mastitis, cattle, bacteria, multiplex PCR

## Abstract

Mastitis is a common mammary gland disease of dairy cattle caused by a wide range of organisms including bacteria, fungi and algae. Mastitis contributes to economic losses of dairy farms due to reduced yield and poor quality of milk. Since the correct identification of pathogens responsible for the development of mastitis is crucial to the success of treatment, it is necessary to develop a quick and accurate test to distinguish the main pathogens causing this disease. In this paper, we describe the development of a test based on the multiplex polymerase chain reaction (PCR) method allowing for the identification of *Streptococcus agalactiae*, *Streptococcus dysgalactiae*, *Streptococcus uberis* and *Staphylococcus aureus.* When creating our test, we relied on the results from new generation sequencing (NGS) for accurate determination of species affiliation. The multiplex PCR test was verified on 100 strains including veterinary samples, ATCC and Polish Collection of Microorganisms (PCM) reference strains. The obtained results indicate that this test is accurate and displays high specificity. It may serve as a valuable molecular tool for the detection of major mastitis pathogens.

## 1. Introduction

Mastitis is the most common disease that affects cattle and buffalo worldwide, both in developing and developed countries and in different climatic zones, which decreases milk production leading to economic loss for dairy farmers [1,2,3]. It has been shown that mastitis is the most frequently diagnosed disease in dairy cows, affecting 25% of cows in the herd annually [4]. Recent research on the costs of treating clinical mastitis in 37 dairy farms showed that the cost per animal is around USD 200 [5].

A variety of organisms, including bacteria, mycoplasmas and algae, have been reported to be associated with the development of mastitis [6]. Pathogens that most commonly cause mastitis can be categorized into two groups: contagious pathogens that spread from cow to cow and environmental pathogens. Contagious pathogens include bacteria like *Staphylococcus aureus*, *Streptococcus agalactiae* and *Corynebacterium bovis*. Environmental pathogens include *Escherichia coli*, *Streptococcus uberis*, *Streptococcus dysgalactiae* and other Gram-positive and catalase-negative cocci [7]. Mastitis can be classified into clinical mastitis (CM) and subclinical mastitis (SCM). CM is characterized by visible abnormalities, such as red, swollen udders and systemic fever in cattle. In the case of SCM, there are no visible irregularities in the udder or milk. However, it is observed that as the somatic cell count (SCC) increases, milk production decreases [8]. The occurrence of mastitis in cattle breeding requires treatment, with antibiotic therapy still being the main therapeutic strategy. Antibiotics are administered in the form of intramammary infusion, intramuscular or intravenous injections [9]. In response to the escalating issue of antimicrobial resistance, both national and international regulations have been put in place to curtail the unnecessary use of antibiotics [10]. It is important to note that not all intramammary infections (IMIs) need antibiotic treatment. The decision to administer antibiotics should be based on diagnostic and sensitivity test outcomes, rather than on presumptive treatment [11].

Various diagnostic methods were developed to determine IMIs, but traditional methods rely on counting the number of somatic cells in the milk and include, e.g., the California mastitis test, Somatic Cell Count (SCC) and electrical conductivity test [12]. Microbiological methods consisting of bacterial isolation and culturing on different media allow us to determine the presence of specific pathogens responsible for the development of the disease. Unfortunately, the main disadvantage of these methods is the time it takes to get the results. Laboratory work related to microbiological diagnostics can take up to several days. In recent years, molecular methods, especially polymerase chain reaction (PCR), have gained popularity [13]. According to Borelli et al., only some Scottish farmers use microbiology laboratories, not because of the cost of the test, but because of the waiting time for the results [14].

In comparison to classical microbiological diagnosis, PCR-based methods are rapid, economic and sensitive. Studies by Bexiga et al. showed that microbiological methods allow the identification of only 47% of mastitis cases compared to the PCR method [15]. Moreover, traditional methods, such as the Somatic Cell Count (SCC), offer quick, cost-effective and field-friendly solutions, but they do not provide specific detection [11].

Another method that can be used to genotype pathogens causing mastitis is next-generation sequencing (NGS). Anis et al. described a method based on targeted NGS using 198 primers which were designed to target 43 bovine pathogens. In comparison to metagenomic sequencing, targeted NGS provides better specificity and sensitivity and lower cost. On the other hand, new pathogens cannot be detected with this method. However, it should be borne in mind that sequencing requires specialized laboratory equipment, properly trained staff and thorough bioinformatics analysis [16].

An effective mastitis control program hinges on the prompt identification of the infection. This can be achieved by comprehending the disease’s progression, developing novel sensitive tests for early screening and implementing management practices to minimize the likelihood of spreading the infection from infected to healthy quarters. Early mastitis detection and pinpointing the responsible agent are vital for its control and treatment. These steps are key in cutting costs, reducing milk yield and quality losses and improving the recovery rate of infected animals [17].

This study aimed to develop a multiplex PCR method, based on NGS data, that could expedite the simultaneous detection and differentiation of *S. agalactiae*, *S. dysgalactiae*, *S. aureus* and *S. uberis.* These pathogens are frequently responsible for mastitis, resulting in both acute and chronic udder infections, which in turn lead to substantial economic losses within the dairy industry. Some species of staphylococci and streptococci are zoonotic pathogens, meaning they can be transmitted between cows and humans, posing a potential risk to human health [18]. Therefore, it is necessary to develop fast, inexpensive and effective tests for identification of these pathogens.

## 2. Materials and Methods

### 2.1. Bacterial Strains

A total of 100 bacterial strains representing different species of genera *Streptococcus*, *Aerococcus*, *Enterococcus* and *Staphylococcus* were used. It included 6 ATCC reference strains (Manassas, VA, USA), 11 PCM strains (Wroclaw, Poland) and 83 bacterial strains derived from veterinary samples (see Appendix A). The strains were obtained from milk samples received from cows with mastitis symptoms. The additional 12 strains isolated from cats and dogs from different infected areas such as ears, skin, larynx, eye, nose, foreskin, urinary tract, bronchial tree and various types of wounds were used. Milk samples and bacteria cultures were transported at a temperature of 2–10 °C. A single colony from bacteria cultures on plates or 100 μL of each milk sample was propagated on CHROMagar™ Mastitis GP medium (CHROMagar™, Paris, France). Then, the metallic blue colonies were grown in parallel on Bile Esculin Azide (Biomaxima, Lublin, Poland) to exclude that the strain belongs to the *Enterococcus* species. The plates were incubated at 37 °C for 18 h ± 2 h. In addition, a 6.5% NaCl tolerance test was performed. Strains selected as *Streptococcus* were incubated for 7 days in 5 mL of brain heart infusion broth (BHI) with the addition of 6.5% NaCl and pH color indicator bromophenol blue at 37 °C. If turbidity or turbidity and a color change of the medium from violet to yellow were observed, belonging of the strain to *Streptococcus uberis* was excluded. Light blue tiny colonies that grew on Chromagar Mastitis GP medium were tentatively typed as *Streptococcus dysgalactiae*/*agalactiae*. Selected colonies were subjected to API Strep 20 biochemical tests (bioMérieux, Warsaw, Poland). Pink colonies that grew on Chromagar Mastitis GP medium were transferred to Chapman (Biomaxima, Lublin, Poland) and Columbia LAB-AGAR media (Biomaxima, Lublin, Poland). Ifthe colonies were yellow and round on Chapman medium and capable of β-haemolysis on Columbia LAB-AGAR, samples were pre-qualified as *S. aureus.* However, when the colonies on Chapman medium were pink or yellow but α- or δ-haemolysis was observed on Columbia LAB-AGAR, the bacteria were still considered as *Staphylococcus* sp. Then, the selected strains were subjected to API Staph biochemical tests. Based on microbiological analyses (Figure 1), the tested environmental strains were initially classified into the species and strains.

### 2.2. Extraction of Bacterial Genomic DNA

Bacterial DNA was extracted from each strain cultured on TSB agar using a Wizard Genomic DNA Purification Kit (Promega, Madison, WI, USA) according to the manufacturer’s instructions. The concentration and quality of isolated DNA was determined using a BioSpectrometer (Eppendorf, Hamburg, Germany) and stored at −20 °C until use.

### 2.3. Next Generation Sequencing

The DNA was quantified using a Qubit Fluorometer with the Qubit^TM^ double-stranded DNA (dsDNA) high-sensitivity (HS) Assay Kit (Invitrogen, Warsaw, Poland). Bacterial genomes were sequenced by NGS. The paired-end genome sequencing library was generated with the Illumina DNA Prep (Illumina Inc., San Diego, CA, USA) according to the manufacturer’s instructions. NGS was performed on the Illumina MiniSeq Instrument (Illumina Inc., San Diego, CA, USA) to generate 2 × 150 bp reads and assumed coverage 150 times.

### 2.4. Bioinformatic Analysis of NGS Data

NGS data were analysed by the Tormes 1.3.0 pipeline [19]. The taxonomy was predicted by K-mare analysis by Kraken2 software [20] and analysis of 16S RNA by RDP Classifier [21]. The taxonomy predictions made by K-mer and 16S RNA methods were validated by Average Nucleotide Identity calculation (ANI) which is a ‘gold standard’ for genus and species-level taxonomy qualification of microorganisms [22]. ANI for all assembled genomes were analyzed by PyANI software [23]. The reference assemblies of genomes with verified taxonomic status used for ANI calculation were downloaded from the NCBI GeneBank collection of assemblies (see Appendix A) [24].

### 2.5. Multiplex PCR Primer Design

For designing PCR test to detect pathogenic bacteria species belonging to *Streptococcus agalactiae*, *Streptococcus dysgalactiae*, *Streptococcus uberis* and *Staphylococcus aureus*, all genomes’ assemblies of family *Streptococcaceae* (16,619 assemblies) and *Staphylococcaceae* (18,820 assemblies) were downloaded from the NCBI assembly database (https://www.ncbi.nlm.nih.gov/assembly, accessed on 30 May 2022). The conservative core regions for the chosen species were detected by RUCS 1.0 software [25]. All downloaded genomes for target species were analyzed against the true negative database consisting of one assembly per species for all species belonging to the genus *Streptococcus* or *Staphylococcus* and without a target species. The detected unique core sequences were analyzed by Primer-BLAST (BLAST+ v2.13.0, Bethesda, USA, https://www.ncbi.nlm.nih.gov/tools/primer-blast/, accessed on 20 June 2022) to design specific primers. After that, the primers were manually curated to achieve proper product sizes and physicochemical properties, which were analyzed by OligoCalc [26] and PrimerDimer [27]. In silico PCRs were performed by Ugene software [28]. The results of isPCR made on 16,619 assemblies of *Streptococcaceae* and 18,820 assemblies of *Staphylococcaceae* were analyzed by in-house-made Python script. The location of the detected regions was analyzed by SnapGene Viewer® software (v5.0, Boston, MA, USA, www.snapgene.com). All primers were synthesized by Eurofins (Eurofins Genomics Germany GmbH, Ebersberg, Germany).

### 2.6. Single PCR Conditions

PCR for each target bacteria, *Streptococcus agalactiae*, *Streptococcus dysgalactiae*, *Streptococcus uberis* and *Staphylococcus aureus*, was optimized separately. The PCR reaction was performed in a mixture of 50 µL:25 µL Color Taq PCR Master Mix, 1 µL of 10 µM each primer and a minimum of 7 ng of bacterial DNA sample plus sterilised water (up to 50 µL). To obtain the optimal conditions of the PCR, several experiments were carried out, with annealing temperatures ranging from 50 °C to 60 °C. The reaction conditions were as follows: 1 cycle of 95 °C for 5 min followed by 30 cycles of 95 °C for 30 s, 50 °C–60 °C for 30 s and 72 °C for 1 min and a final extension step of 72 °C for 7 min. Amplification reactions were carried out with a Color Taq PCR Master Mix (EurX, Gdansk, Poland) in a Nexus Gradient thermocycler (Eppendorf, Hamburg, Germany). The PCR products were analyzed by electrophoresis on 1.5% to 2.5% agarose gels in TAE buffer for 1 h 10 min to 1 h 45 min at 32 V and stained by the nucleic acid stain (SimplySafe™, Eurx). The results were read using the Essential V6 imaging system (UVItec Limited, Cambridge, UK).

### 2.7. Multiplex PCR Conditions

To evaluate the accuracy of multiplex PCR for species identification, 17 reference strains, 71 strains derived from subclinical mastitis cases and 12 from different animals were used in this study. The multiplex PCR was performed by mixing the four primer pairs in optimized annealing temperatures and optimized concentrations of upstream and downstream primers. The PCR reaction was performed in a mixture of 50 µL:25 µL Color Taq PCR Master Mix, 1 µL of 10 µM each primer and a minimum of 7 ng/of bacterial DNA sample plus sterilised water (up to 50 µL). The reaction conditions were as follows: 1 cycle of 95 °C for 5 min; followed by 30 cycles of 95°C for 30 s, 50 °C for 30 s and 72 °C for 1 min; and a final extension step of 72 °C for 7 min. Amplification reactions were carried out with a Color Taq PCR Master Mix (EurX, Gdansk, Poland) in a Nexus Gradient thermocycler (Eppendorf, Hamburg, Germany). The PCR products were analyzed by electrophoresis on 2.5% agarose gels in TAE buffer for 1 h 45 min at 32 V and stained by the nucleic acid stain (SimplySafe™, Eurx). The results were read using the Essential V6 imaging system (UVItec Limited, Cambridge, UK).

## 3. Results

### 3.1. Taxonomy Analysis of NGS Data Analysed Bacterial Strains

The results of taxonomy analysis based on bioinformatics NGS data analysis are presented in Appendix A. After the bioinformatic analysis, 9 strains changed taxonomy classification at the species level, 3 strains were reclassified at the genus level and in cases of 10 unclassified *Staphylococcus*, strains were assigned to the species level.

### 3.2. Primer Design

Detection of conservative regions by RUCS software allowed for the design of specific primers for the detection of chosen bacterial species. The physicochemical properties of primers are presented in Table 1 and Table 2. The size of the PCR products was designed to allow for easy eye detection on an agarose gel. The designed primers were checked for the creation of homo- and heterodimers which can drastically decrease the efficiency of the multiplex PCR. The results of this test are presented in Appendix A. These results show the low affinity between primers and the low probability of the creation of homo- and heterodimers.

The designed PCR test was first evaluated by in silico PCR. The results of isPCR, presented in Appendix A, show high scores for sensitivity, specificity, precision and accuracy parameters on the tested dataset consisting of taxonomically correct genome assemblies of *Streptococcaceae* and *Staphylococcaceae* (Appendix A: the list of assemblies used to isPCR).

### 3.3. Establishment of Single PCR Conditions

We determined the conditions of a single PCR using primers, shown in Table 2. To determine the optimum annealing temperature for a single PCR, temperatures ranging from 50 °C to 60 °C were tested. Based on the obtained results we have selected the annealing temperature at which the most satisfactory PCR products for all of the primers were received. Amplicons received in multiplex PCR must differ in length at least 40–50 bp from each other so that they are distinguishable after agarose gel electrophoresis. The Sraga1_F/R, Srdys1_F/R, Srube3_F/R and Staur3_F/R primers resulted in product sizes of 130 bp, 66 bp, 272 bp and 362 bp, respectively.

### 3.4. Multiplex PCR

After determining the optimal reaction conditions for all primers, we performed the multiplex PCR for 96 samples. Results of multiplex PCR are shown in Figure 2, Appendix A and Table 3. The results indicate that the primers in both single and multiplex PCR assays amplified the size-specific products corresponding to bacterial strains.

The multiplex PCR test was validated on both ATCC and PCM reference strains and strains derived from veterinary samples. The multiplex PCR results are consistent with subsequent bioinformatic analyses of samples after genome sequencing. Comparing the results of microbiological analyses with those obtained by NGS sequencing and multiplex PCR, it was noted that 22 of 52 sequenced strains were misclassified after microbiological tests.

## 4. Discussion

Mastitis is a major challenge in dairy cattle breeding worldwide, where pathogens remain the same but their prevalence varies between regions and continents. Therefore, a rapid and precise diagnostic test is necessary for early detection of pathogenic bacteria, surveillance and prevention of mastitis. Fast and accurate diagnostics allow for the implementation of appropriate treatment and reduction of costs associated with the occurrence of the disease. Traditional methods of bacteria identification rely on microbiological analyses; however, they are not as accurate as we would expect them to be. In daily work, the chromogenic culture media have been implemented for veterinarians to be used on farms to expedite treatment decisions. Chromogenic culture media are a quick method of differentiating bacterial species/groups by colony colour. They have shown high sensitivity and specificity for identifying *S. aureus* in clinical mastitis milk samples. They can identify *E. coli*, *Klebsiella* spp. and *Enterobacter* spp. as well as *Streptococcus agalactiae*, *Streptococcus uberis* and *Streptococcus dysgalactiae.* According to the literature, chromogenic culture media could be an alternative for rapid identification of subclinical mastitis, but the observer experience is crucial for proper identification [11]. In our initial microbiological testing, we were using CHROMagar Mastitis GP chromogenic medium to be able to differentiate *S. aureus* and the above mentioned streptococci strains. While the pink *S. aureus* was different from other colonies and easy to differentiate, streptococci strains were all in the blue colour with different shades. *S. uberis* was morphologically similar to *Enterococcus* sp. colonies, thus without inoculation on Bile Esculin Azide, a differential media for *Enterococcus* spp., the visual identification was impossible. Moreover, colonies of *S. agalactiae* and *S. dysgalactiae* were smaller and brighter than *S. uberis/Enterococcus* sp. but also similar to each other and *Aerococcus* sp., so only genetic analysis could distinguish them. We also conducted an additional liquid test with culturing bacteria in Brain Heart Infusion broth supplemented with 6.5% NaCl for differentiation between *Streptococcus* sp. and *Aerococcus/Enterococcus* species, but it is a very time-consuming method due to a long incubation period of about 7 days. Hence, we decided to confirm the results with API tests, but they were usually below 90% of the identification rate in the case of API 20 Strep tests. In the case of *S. aureus* identification by API Staph test the %ID was >90%, however, there was a note that it could be *S. pseudintermedius* if the sample was of veterinary origin. While this method is generally considered relatively fast, there are bacterial strains for which API test results can be inconclusive, leading to their classification at the genus level only, which was unsatisfactory for our purposes. Although microbiological methods can serve as on-farm mastitis tests for identifying *S. aureus*, they are limited to specifying streptococci only at the genus level. In contrast, genetic techniques, such as our multiplex PCR, allow precise identification of the exact species. Moreover, the interpretation of results obtained as colours on the API strip or as bacteria morphology on the plate is highly subjective, while PCR result interpretation is relatively objective. Hence, we decided to design a rapid and accurate multiplex PCR for the differentiation of the four main bacterial species causing mastitis: *Streptococcus agalactiae*, *Streptococcus dysgalactiae*, *Staphylococcus aureus* and *Streptococcus uberis*. The primer design step is crucial in creating multiplex PCR reactions, thus primers designed for this test were analysed by in silico PCR on 16,619 sequences to calculate the test statistics (see Appendix A).

The diagnostic test described in this work was verified on 100 samples. In addition to the typical mastitis strains, we also included additional strains from companion animals. Among these, *Staphylococcus* species serve as opportunistic pathogens in various skin infections, such as pyoderma in dogs. The most prevalent pathogens associated with pyoderma in immunocompromised dogs and skin infections in other companion animals include *S. aureus* and *S. pseudintermedius* [29,30]. These samples were treated as another control for our test as, according to our NGS data, strains classified as *S. pseudintermedius* were negative in our PCR test for *S. aureus*, thus, we were able to verify the performance of our PCR test for *S. aureus* strains where the API results were inconclusive.

In addition, some of the tested bacterial DNA samples were subjected to next-generation sequencing to check the accuracy of the test (see Appendix A). To the best of our knowledge, it is one of the widest analyses ever made to design diagnostic tests for mastitis pathogens.

It is valuable to mention that in the context of bovine mastitis, *Staphylococcus aureus* presents a significant challenge as a facultative intracellular pathogen. It can thrive both inside and outside host cells, making it extremely difficult for antibiotics to eliminate the residing pathogen. Eukaryotic cells act as an additional barrier, hindering drug penetration. Once inside the cell, *S. aureus* can survive and multiply within the acidic environment of the cellular phagolysosome. Subsequently, it can escape from the phagolysosome, leading to cell death and the release of the pathogen. *S. aureus* is also capable of forming biofilms and developing microabscesses inside the mammary gland. This phenomenon contributes to chronic, persistent infections and further spread of the pathogen within the herd [31,32]. Epithelial cells from the mammary gland are naturally present in milk, albeit in small quantities that can vary among cows. If these cells are infected, then it is possible to isolate bacterial DNA. Our research suggests that our test could directly detect *Staphylococcus aureus* from a milk sample without the need for microbiological diagnostics. We could utilize the same isolation protocol or protocols similar to the one employed by Pokorska et al. [33]. This approach would significantly shorten the time required for screening. However, in this particular case, our primary objective was to isolate bacteria and preserve them for future investigations.

It has to be noted that mastitis can be caused by a variety of microorganisms, thus it is common to encounter mixed infections. According to the National Mastitis Council, samples with two identified pathogens fall under the definition of a mixed infection. However, if more than two pathogens are present, the samples are typically described as contaminated—unless a colony of a cow-associated microorganisms (such as *S. aureus*, *S. agalactiae*, *S. dysgalactiae*, or *T. pyogenes*) is detected [34]. We did not specifically test whether our diagnostic test would detect mixed infections because our primary focus was on isolating bacterial monocultures. We compared microbiological and biochemical methods with multiplex PCR to validate our diagnostic results and shorten the screening process time. However, in the case of *Staphylococcus* sp. 043PP2022, we detected the presence of both *S. aureus* and *S. uberis.* Interestingly, during microbiological analyses, we initially considered this strain as a pure monoculture. This observation indicates minimal contamination by another strain. Consequently, it suggests that our test is sensitive enough to detect lower amounts of DNA than we initially assumed. This case underscores the potential of our test to handle the detection of mixed infections—an intriguing avenue for future development.

As the fast and effective diagnosis of pathogens causing mastitis is still a challenge, new methods that allow the accurate identification of bacterial species are being constantly sought [35,36]. The most popular and traditional test used for early mastitis screening is the California mastitis test which involves taking a milk sample directly from each teat and placing it on the four fields of a paddle. The added reagent changes the consistency of the milk to a gel when the somatic cell number is increased. It is quick, simple and inexpensive, but often unavailable for farmers to buy [37]. However, this method only determines the presence of ongoing infection but does not allow for the identification of the pathogen, therefore further diagnostics such as our multiplex PCR are necessary to implement appropriate antibiotic therapy.

In the literature there is an assay based on lateral flow developed by Sayed et al. who used rabbit polyclonal antibodies specific to the antigen of five selected mastitis-causing agents (*S. aureus*, *E. coli*, *K. pneumoniae*, *S. agalactiae*, *S. pyogenes*). This kit was applied for bovine milk examination and its sensitivity and specificity in comparison to standard microbiological tests were 80.83% and 90.53%, respectively. The results can be obtained in 10 min, but without 6 h in 37 °C pre-incubation of milk sample in nutrient media, the sensitivity of the test is significantly decreased [38]. Deb et al. created an LFA test specific to *S. aureus* detection in milk with the use of stable colloidal silver nanoparticles derived from mango leaves as a conjugate with recombinant SpA protein from *S. aureus*. This test lasts for 7–8 h (if the enrichment step of the milk sample with selective media is not included) and can detect bacteria at a very low level of 1000 CFU/mL of milk sample. Moreover, the test achieved accuracy, specificity and sensitivity levels of 97.39%, 98.03% and 96.1%, respectively [39]. This test offers a great opportunity for quick identification of mastitis, but only in the presence of *S. aureus*, while other mastitis-causing pathogens will not be detected. Dobrut et al. also developed an LFA diagnostic test for *S. uberis*, *S. agalactiae* and *E. coli* based on the detection of species-specific surface proteins. The proteins showed little to no reaction with negative samples. Any observed reactivity might be due to the animals’ prior exposure to the bacteria under study. Despite the absence of these bacteria in milk samples and the lack of infection symptoms, some reactivity was still observed [35]. In comparison, our multiplex PCR has better parameters than the abovementioned tests of about 99.65%, 98.83% and 99.92% for accuracy, specificity and sensitivity for *S. aureus*, respectively. Moreover, it allows for the identification of *S. aureus*, *S. dysgalactiae*, *S. uberis* and *S. agalactiae* species, none of which are identified in the abovementioned tests in this configuration.

The literature suggests various immunoassay tests for early inflammation screening utilizing biomarkers like milk amyloid A [40], cathelicidins [41], myeloperoxidase enzyme of milk neutrophils [42], IL-6 [43] as well as lateral flow assays for diagnosing mastitis [44]. While the abovementioned methods offer relatively low costs and high precision in comparison to standard diagnostic methods, we cannot obtain results indicating the bacteria species, which is crucial for the implementation of appropriate antibiotic therapy, thus it can be used as a screening test, but still, microbiological and genetic tests should be conducted.

To sum up, traditional and immunological tests are focused on detecting inflammation or a narrow range of bacterial species in the udder, while the species selection depends mainly on the region and prevalence. The epidemiological data show that the mastitis-causing bacteria prevalence between countries is significantly different. Our multiplex PCR has the advantage of detecting four different species and can be easily modified by adding primers to detect additional species of interest making it a universal test. Many researchers analysing mastitis pathogens rely on the PathoProof™ kit from ThermoFisher, which identifies 15 different species [45]. However, this kit was developed and validated on strains whose taxonomic classification was confirmed by the 16S rRNA method, which in recent years has proven to be unsuitable for determining species-level classification [46]. In our test, species classification was analysed by NGS and we calculated ANI (Average Nucleotide Identification) relative to the declared species.

It has to be noted that both PCR and immunological tests require time between the submission and report from the study as well as a laboratory equipped with specialized devices and staff, unless the immunological test is designed in such a way that it can be used on-farm by a farmer or veterinarian [10].

According to the microbiological diagnostic scheme (as shown in Figure 1), the execution time varies from 4 to 7 days. The longer duration for *Streptococcus* identification is due to the utilization of a salt tolerance test. However, by employing multiplex PCR and streamlining certain steps on differentiation media, we can significantly reduce the time needed to obtain results to 3 days. Specifically, we can achieve this by cultivating bacteria on CHROMagar Mastitis GP and selecting and inoculating suspicious colonies on enrichment media (e.g., TSA) for further DNA isolation and genetic analysis. The cost comparison between full microbiological diagnostics (including biochemical API tests) and the shortened approach with multiplex PCR show that expenses remain equivalent for *Staphylococcus*. However, when it comes to *Streptococcus*, the genetic diagnostic approach proves to be 27% cheaper.

It is to be noted that a possible limitation of this test is that it will not discriminate between dead or living cells, although the goal was to connect microbiological analyses with genetics and to keep detected bacteria alive, thus we have not been implementing quantitative PCR (qPCR). Nonetheless, qPCR will show us the relative abundance of targeted bacteria in the sample, which has an advantage in assessing the stage of an infection. We opted for standard PCR due to its simplicity, making it accessible even for less experienced researchers. Additionally, the lower costs associated with standard PCR, including the availability of laboratory equipment and reagents, further support its implementation compared to qPCR. The primers we designed have not been validated for qPCR and the differences in product sizes were intentionally selected to allow easy differentiation on agarose gels. However, these differences enable discrimination on melting curves in qPCR as well, thus, they can be easily utilized in qPCR in future activities.

## 5. Conclusions

Various technologies are available for tracking mastitis and IMI in dairy cows using milk as a diagnostic medium, but their effectiveness in managing udder health is still being studied. Farmers and veterinarians should utilize these technologies more efficiently, while PCR tests offer advantages over other methods, as they can be easily replicated and are more objective in interpreting results. Our multiplex PCR test, developed based on recent taxonomic classifications and validated through next-generation sequencing, is effective in detecting udder infections and can be easily modified to detect a plethora of bacterial species. Furthermore, our PCR test can be upgraded to qPCR utilizing the same primers to enhance its diagnostic value. The duration of diagnostics is shortened, especially for *Streptococcus* sp. identification together with cost reduction in comparison to classical microbiological and biochemical diagnostics. Moreover, our PCR test shows potential for the detection of mixed infections. However, human supervision is still essential for accurately interpreting results and reducing the risk of misdiagnosis.

## Figures and Tables

**Figure 1 pathogens-13-00423-f001:**
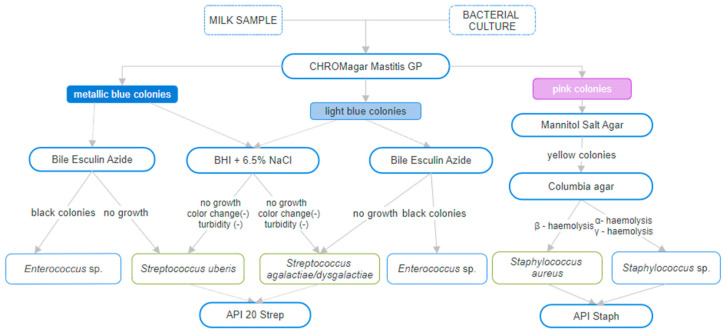
Scheme of microbiological diagnostics of milk samples and bacterial cultures.

**Figure 2 pathogens-13-00423-f002:**
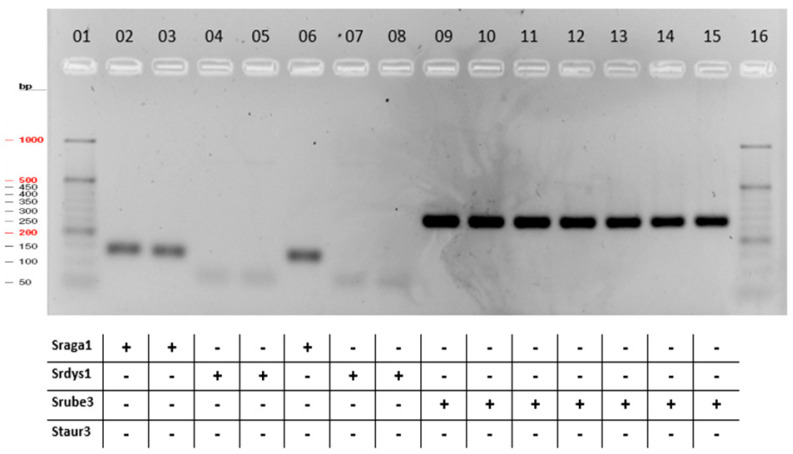
Multiplex PCR results. Lane 01 and 16, 50 bp DNA ladder (with the following distribution—50, 100,150, 200, 250, 300, 400, 500, 600, 700, 800, 900, 1000); Lane 02, *Streptococcus agalactiae* 023PP2021; Lane 03, *Streptococcus agalactiae* 024PP2021; Lane 04, *Streptococcus agalactiae* 025PP2021; Lane 05, *Streptococcus agalactiae* 026PP2021; Lane 06, *Streptococcus agalactiae* 027PP2021; Lane 07, *Streptococcus dysgalactiae* 001PP2016; Lane 08, *Streptococcus dysgalactiae* 002PP2016; Lane 09, *Streptococcus dysgalactiae* 003PP2016; Lane 10, *Streptococcus uberis* 041PP2021; Lane 11, *Streptococcus uberis* 045PP2021; Lane 12, *Streptococcus uberis* 059PP2021; Lane 13, *Streptococcus uberis* 067PP2021; Lane 14, *Streptococcus uberis* 071PP2021; Lane 15, *Streptococcus uberis* 103PP2021.

**Table 1 pathogens-13-00423-t001:** Basic information of designed primers including the name of detected species, primers sequence, the PCR product length and the genome region detected by the primers, gene ID of detected conservative region for targeted species.

Species	Primer	Sequence	Product [nt]	Detected Conservative Region	GenBank ID
*S. agalactiae*	F	CTGTGCTTAGTCCACTTGAA	130	zinc ABC transporter substrate-binding protein AdcA	66885516
	R	GGAGCTACTTCTTTACCTGC			
*S. dysgalactiae*	F	CTTGCTCCTTTGAAGTAATCA	66	CDP-diacylglycerol—glycerol-3-phosphate 3-phosphatidyltransferase	83689676
	R	TGTCTTTCTCTATGTTGCTCT			
*S. uberis*	F	CGTGAAGATGAAGATGTCCTA	272	Two genes encoding: YitT family protein and ABC-F family ATPase	58021974, 58021975
	R	GTGGGTTCAATGTCTCCAG			
*S. aureus*	F	CCTTGACTCGCAATGTTAAG	362	phage infection protein	3921461
	R	ATTGAAGAAAATGTGCCGAC			

**Table 2 pathogens-13-00423-t002:** Detailed information about designed primer length, melting temperature calculated by basic method, the GC count, the self-complementarity, the self-complementarity of primers 3’ ends and the ability to create hairpins.

Species	Primer	Length [nt]	Tm (basic) [°C]	GC [%]	Self	Self-3′	Hairpin
*S. agalactiae*	F	20	49.7	45	4	3	No
	R	20	51.8	50	4	2	No
*S. dysgalactiae*	F	21	48.5	38.1	4	2	No
	R	21	48.5	38.1	2	0	No
*S. uberis*	F	21	50.5	42.86	2	2	No
	R	19	51.1	52.63	3	2	No
*S. aureus*	F	20	49.7	45	6	6	No
	R	20	47.7	40	3	3	No

**Table 3 pathogens-13-00423-t003:** Multiplex PCR results.

	Primers			
Sample	Sraga1_F/R	Srdys1_F/R	Srube3_F/R	Staur3_F/R
*Staphylococcus warneri* ATCC 27836				
*Staphylococcus epidermidis* ATCC 14990				
*Streptococcus dysgalactiae* ATCC 12394		+		
*Aerococcus viridans* ATCC 11563				
*Enterococcus faecalis* ATCC 29212				
*Staphylococcus aureus* ATCC 6538P				+
*Staphylococcus aureus* PCM 2267				+
*Staphylococcus aureus* PCM 458/2195				+
*Staphylococcus aureus* PCM 2101				+
*Staphylococcus aureus* PCM 2054				+
*Staphylococcus aureus* PCM 1650				+
*Staphylococcus aureus* PCM 1116				+
*Staphylococcus aureus* PCM 1115				+
*Staphylococcus aureus* PCM 1102				+
*Staphylococcus aureus* PCM 565				+
*Staphylococcus aureus* PCM 502				+
*Staphylococcus aureus* PCM 1937				+
*Streptococcus agalactiae* 023PP2021	+			
*Streptococcus agalactiae* 024PP2021	+			
*Streptococcus dysgalactiae* 004PP2021		+		
*Streptococcus dysgalactiae* 005PP2021		+		
*Streptococcus agalactiae* 027PP2021	+			
*Streptococcus dysgalactiae* 001PP2016				
*Streptococcus dysgalactiae* 00PP2016		+		
*Streptococcus uberis* 120PP2022			+	
*Streptococcus uberis* 041PP2021			+	
*Streptococcus uberis* 045PP2021			+	
*Streptococcus uberis* 059PP2021			+	
*Streptococcus uberis* 067PP2021			+	
*Streptococcus uberis* 071PP2021			+	
*Streptococcus uberis* 103PP2021			+	
*Streptococcus uberis* 105PP2021			+	
*Streptococcus uberis* 112PP2021			+	
*Aerococcus urinaeequi* 001PP2021				
*Aerococcus urinaeequi* 002PP2021				
*Mammaliicoccus sciuri* 004PP2021				
*Enterococcus faecium* 003PP2021				
*Enterococcus faecium* 004PP2021				
*Enterococcus faecalis* 022PP2021				
*Staphylococcus haemolyticus* 004PP2022				
*Staphylococcus sciuri* 001PP2020				
*Streptococcus agalactiae* 028PP2022	+			
*Staphylococcus equorum* 003PP2022				
*Staphylococcus equorum* 001PP2022				
*Staphylococcus equorum* 002PP2022				
*Mammaliicoccus vitulinus* 001PP2022				
*Enterococcus* sp.009PP2022				
*Enterococcus* sp. 010PP2022				
*Enterococcus* sp. 011PP2022				
*Enterococcus* sp. 012PP2022				
*Enterococcus* sp. 013PP2022				
*Staphylococcus* sp. 008PP2022				+
*Staphylococcus* sp. 009PP2022				+
*Staphylococcus* sp. 010PP2022				
*Staphylococcus* sp. 011PP2022				+
*Staphylococcus* sp. 012PP2022				
*Staphylococcus* sp. 013PP2022				+
*Staphylococcus* sp. 014PP2022				+
*Staphylococcus* sp. 015PP2022				+
*Staphylococcus* sp. 016PP2022				+
*Staphylococcus* sp. 017PP2022				
*Staphylococcus* sp. 018PP2022				+
*Staphylococcus* sp. 019PP2022				+
*Staphylococcus* sp. 023PP2022				
*Staphylococcus* sp. 025PP2022				+
*Staphylococcus* sp. 026PP2022				
*Staphylococcus* sp. 027PP2022				
*Staphylococcus* sp. 030PP2022				
*Staphylococcus* sp. 031PP2022				
*Staphylococcus* sp. 032PP2022				
*Staphylococcus* sp. 034PP2022				
*Staphylococcus* sp. 038PP2022				
*Staphylococcus* sp. 039PP2022				
*Staphylococcus* sp. 040PP2022				
*Staphylococcus* sp. 041PP2022				
*Staphylococcus* sp. 042PP2022				
*Staphylococcus* sp. 043PP2022			+	+
*Staphylococcus* sp. 044PP2022				+
*Staphylococcus* sp. 045PP2022				
*Staphylococcus* sp. 046PP2022				
*Staphylococcus* sp. 057PP2022				+
*Mammalicoccus sciuri* 005PP2022				
*Streptococcus agalactiae* 021PP2018	+			
*Streptococcus agalactiae* 022PP2021	+			
*Streptococcus* sp. 003PP2017			+	
*Streptococcus* sp. 004PP2017			+	
*Streptococcus* sp. 006PP2017				
*Streptococcus* sp. 008PP2018		+		
*Streptococcus* sp. 009PP2019			+	
*Streptococcus* sp. 014PP2022			+	
*Streptococcus* sp. 017PP2022				
*Streptococcus* sp. 018PP2022				
*Staphylococcus pseudintermedius* 001PP2023				
*Staphylococcus pseudintermedius* 002PP2023				
*Staphylococcus simulans* 002PP2022				
*Staphylococcus pseudintermedius* 003PP2023				
*Staphylococcus pseudintermedius* 004PP2023				
*Staphylococcus pseudintermedius* 005PP2023				
*Staphylococcus pseudintermedius* 006PP2023				
*Staphylococcus pseudintermedius* 007PP2023				
*Staphylococcus pseudintermedius* 011PP2023				
*Staphylococcus pseudintermedius* 012PP2023				
NTC (No Template Control)				

+ gene amplification (band present on the gel).

## Data Availability

All genomes assemblies used to design this multiplex PCR test are available in https://www.ncbi.nlm.nih.gov/assembly/ at the NCBI RefSeq assembly ID included in the Appendix A. The original contributions presented in the study are included in the article/Appendix A, further inquiries can be directed to the corresponding author/s.

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
