# Peer review of "A Rapid and Inexpensive PCR Test for Mastitis Diagnosis Based on NGS Data"

_pathogens, 2024, doi:10.3390/pathogens13050423_

Round 1

Reviewer 1 Report

Comments and Suggestions for Authors

Good quality overall, I have only detected three limitations:

1. Inclusion of Antibiotic Resistance Genes in PCR Method: The manuscript presents an innovative PCR-based method for identifying pathogens in mastitis. However, incorporating the detection of genes related to antibiotic resistance could significantly enhance the diagnostic utility of this method. By enabling the identification of resistant strains, this addition could facilitate more effective treatment strategies for subclinical mastitis. I recommend the authors discuss this potential enhancement, or consider incorporating it into their method.

2. Advantages of qPCR Over Traditional PCR: The current method employs traditional PCR, but the use of quantitative PCR (qPCR) might offer substantial benefits, such as faster processing and improved multiplexing capabilities. I suggest the authors compare these methodologies in their discussion, highlighting why traditional PCR was chosen and how qPCR could potentially improve their method in future iterations.

3. Detection of Intracellular Infections: The method's ability to identify intracellular infections, specifically those caused by S. aureus in bovine mastitis, isn't addressed. Considering the prevalence and significance of such infections, it would be valuable for the authors to discuss whether their methodology can detect these types of infections and, if not, what modifications might be necessary to do so.

Minor Comments:

- Line 38-39: The sentence structure here is awkward. Clarification or rephrasing could enhance readability.

- Line 41: The word "tool" seems informal in this context. A term like "therapeutic strategy" might be more appropriate.

- Line 42: There's a missing period after reference 9. Please add this for consistency.

- Line 44: A citation seems necessary after the mention of "antibiotics" to support the statement.

- Line 64: The sentence beginning with "Anis et al." is difficult to follow. Simplifying or breaking it into smaller sentences could improve clarity.

- Sections 2.5 onwards: Please ensure that all species and genus names are italicized, adhering to scientific naming conventions. 

Comments on the Quality of English Language

Good quality, only minor issues detected.

Author Response

Dear Reviewer,
Thank you for serving as the Reviewer to our manuscript pathogens-2949471 entitled: "A rapid and 
inexpensive PCR test for mastitis diagnosis based on NGS data". We wish to thank you for your 
supportive comments and constructive suggestions, and are pleased to inform you that in the enclosed 
revised manuscript we have addressed all the concerns which we agreed with. All changes appear in 
revision format in the revised manuscript. Attached please find a detailed point-by-point reply.
We are pleased that the Reviewer found our study in good quality and appreciate all his/her suggestions.
1) Inclusion of Antibiotic Resistance Genes in PCR Method. The manuscript presents an innovative 
PCR-based method for identifying pathogens in mastitis. However, incorporating the detection 
of genes related to antibiotic resistance could significantly enhance the diagnostic utility of this 
method. By enabling the identification of resistant strains, this addition could facilitate more 
effective treatment strategies for subclinical mastitis. I recommend the authors discuss this 
potential enhancement or consider incorporating it into their method.
Our goal in this paper was to develop a PCR test that can distinguish between the species causing 
mastitis. We concentrated on finding universal primers that can be applied, independently from 
antibiotic resistance that can occur. We of course agree with the Reviewer that antibiotic 
resistance in pathogenic strains is a big issue nowadays therefore, in our opinion, it would be 
valuable to identify resistant strains in the next step of analysis to provide effective treatment. 
We will consider it as an enhancement of our method.
2) Advantages of qPCR Over Traditional PCR: The current method employs traditional PCR, but 
the use of quantitative PCR (qPCR) might offer substantial benefits, such as faster processing 
and improved multiplexing capabilities. I suggest the authors compare these methodologies in 
their discussion, highlighting why traditional PCR was chosen and how qPCR could potentially 
improve their method in future iterations.
In general, we agree with the Reviewer that qPCR can be beneficial here although, for our first 
test, we have chosen traditional PCR. It is included in the Discussion, as the Reviewer suggested, 
and we are thinking about enhancement of our test to qPCR in the future.
3) Detection of Intracellular Infections: The method's ability to identify intracellular infections, 
specifically those caused by S. aureus in bovine mastitis, isn't addressed. Considering the 
prevalence and significance of such infections, it would be valuable for the authors to discuss 
whether their methodology can detect these types of infections and, if not, what modifications 
might be necessary to do so.
The Reviewer is right that we did not address these infections. However, we believe that our 
methods can detect intracellular infections as well and we modified our paper to underline that.
All minor comments suggested by the Reviewer are addressed and can be followed in a modified 
manuscript in revision format.
We believe that the revised manuscript is substantially improved, both in style and content, and hope 
that it is now acceptable for publication in the Pathogens Journal. If there are any additional questions, 
please do not hesitate to contact me. We look forward to hearing from you in due time.
Best regards,
Agnieszka Kajdanek

Reviewer 2 Report

Comments and Suggestions for Authors

In the manuscript ID pathogens-2949471, the authors report the development of a novel multiplex PCR protocol able to detect and differentiate four main pathogens responsible for mastitis in dairy cattle. After optimizing the single-target reactions, the multi-target test was validated on a collection of bacterial isolates, including reference strains, isolates recovered from mastitis infections and strains isolated from companion animals, which were previously identified by culture-dependent assays and subjected to sequencing analysis. The authors conclude that their protocol can provide prompt and reliable results in the diagnosis of mastitis in a shorter time compared to routine cultural assays.

The manuscript is interesting, since it underlines a current and actual issue, i.e., the need for implementing the classical culture-based diagnostics with molecular assays able to provide valuable data in a reduced amount of time. The paper is well organized, with an informative introduction, a detailed method section and a proper discussion of the results. Unfortunately, it lacks novelty, and two main weaknesses can be evidenced about the application of the proposed protocol in the diagnosis of mastitis:

·        The developed multiplex PCR was validated on DNA purified from bacterial isolates and not from samples suggesting mastitis infection (e.g., milk); therefore, its efficacy and reliability cannot be evaluated in the case of polymicrobial infections, especially when more target species are present in the same sample;

·        As PCR assay, it has not the ability to differentiate DNA coming from dead cells present in the sample or from live and virulent bacterial cells; moreover, a quantitative assay should be preferred, in order to evidence the different abundance of the detected pathogens. Such assays have already been described in literature (doi: 10.3168/jds.2021-20940; doi: 10.3390/pathogens12070935); the authors should consider modifying their PCR into a qPCR, or at least developing a quantitative molecular assay for one species of interest.

According to this reviewer, the paper needs major revisions and can be considered for publication after addressing the reported concerns.

 MINOR COMMENTS

-Line 79, please correct “These are among the most common pathogens causing mastitis”;

-Lines 140, 141, 158 and 159, please type the bacterial species names in italic.

Author Response

Dear Reviewer,
Thank you for serving as the Reviewer to our manuscript pathogens-2949471 entitled: "A rapid and 
inexpensive PCR test for mastitis diagnosis based on NGS data". We wish to thank you for your 
supportive comments and constructive suggestions, and are pleased to inform you that in the enclosed 
revised manuscript we have addressed all the concerns which we agreed with. All changes appear in 
revision format in the revised manuscript. Attached please find a detailed point-by-point reply.
We are pleased that the Reviewer found our study in good quality and appreciate all his/her suggestions.
1) The developed multiplex PCR was validated on DNA purified from bacterial isolates 
and not from samples suggesting mastitis infection (e.g., milk); therefore, its efficacy 
and reliability cannot be evaluated in the case of polymicrobial infections, especially 
when more target species are present in the same sample. 
Multiplex PCR was indeed validated on DNA from bacterial isolates although they were 
coming e.g. from milk samples received from cows with mastitis symptoms. The idea 
was to shorten the time needed for classical diagnosis, i.e. incubating samples on 
different selective media, performing confirmations like API tests etc. We believe that 
the detection of polymicrobial infections is not excluded e.g. in the case of sample 
Staphylococcus sp. 043PP2022 presented in the manuscript.
2) As a PCR assay, it cannot differentiate DNA coming from dead cells present in the 
sample or from live and virulent bacterial cells; moreover, a quantitative assay should 
be preferred, to evidence the different abundance of the detected pathogens. Such assays 
have already been described in the literature (doi: 10.3168/jds.2021-20940; doi: 
10.3390/pathogens12070935); the authors should consider modifying their PCR into a 
qPCR, or at least developing a quantitative molecular assay for one species of interest.
The reviewer is right here that molecular tests cannot distinguish between dead and 
living cells as DNA can be isolated from both. However, the idea of the test is to connect 
the first part of microbiological analysis, i.e. having bacteria grown on general media 
and then isolation of DNA and PCR test, therefore the preference would be for living 
cells. Another issue concerning qPCR assay was addressed in a Discussion part of a 
revised manuscript. In general, we agree with the Reviewer that having qPCR would be 
beneficial and we are thinking about the enhancement of our test in this direction in the 
future.
All minor comments suggested by the Reviewer are addressed and can be followed in a modified 
manuscript in revision format.
We believe that the revised manuscript is substantially improved, both in style and content, and hope 
that it is now acceptable for publication in the Pathogens Journal. If there are any additional questions, 
please do not hesitate to contact me. We look forward to hearing from you in due time.
Best regards,
Agnieszka Kajdanek

Reviewer 3 Report

Comments and Suggestions for Authors

Dear Authors

I find the work very interesting and valuable. Mastitis is the most common disease of dairy cows, contributing to the economic losses of dairy farms. The correct identification of pathogens is crucial to the success of treatment. It is necessary to develop a quick and accurate test to distinguish the main pathogens causing this disease. A developed multiplex PCR test could be very helpful for effective identification of the pathogens that cause udder infections.

I am not sure that PCR methods are simple, cheap, and quick to identify. It needs a time, and it is combine always microbiological analysis and PCR analysis. Plus for treatment is necessary to find sensitivity of bacterias for antimicrobial drugs. I think this multiplex method is very important for scientist for accurate identification of specific pathogens.

How long does it last for the whole identification? How much would it cost to analyze one sample compared to the other described analyses in discusion?I recommend removing the name of the article, inexpensive.

The introduction and methods are well described. The conclusion should be reformulated and improved.

Minor mistakes

Lines 138-159 Please use italic Streptococcus agalactiae,...

Results

Table 1 is not clear enough, describe it better in the text, e.g. what does mean -, what is not aplicable?

Describe it in legend.

In table 2 and also in other text and figure 3: Are primer abbreviations Sraga_1_F Sraga_1_R needed?  I think it could be written  F and R (or For and Rew)?

Calculations should be replaced from the table and placed to the text or to the legend.

Sincerely

Author Response

Dear Reviewer,
Thank you for serving as the Reviewer to our manuscript pathogens-2949471 entitled: "A rapid and 
inexpensive PCR test for mastitis diagnosis based on NGS data". We wish to thank you for your 
supportive comments and constructive suggestions, and are pleased to inform you that in the enclosed 
revised manuscript we have addressed all the concerns which we agreed with. All changes appear in 
revision format in the revised manuscript. Attached please find a detailed point-by-point reply.
We are pleased that the Reviewer found our study in good quality and appreciate all his/her suggestions.
1) I am not sure that PCR methods are simple, cheap, and quick to identify. It needs time, 
and it combines always microbiological analysis and PCR analysis. Plus for treatment 
is necessary to find the sensitivity of bacteria for antimicrobial drugs. I think this 
multiplex method is very important for scientists for accurate identification of specific 
pathogens.
How long does it last for the whole identification? How much would it cost to analyze 
one sample compared to the other described analyses in discussion? I recommend 
removing the name of the article, inexpensive. 
We believe that compared to traditional diagnosis, which is time-consuming, a test 
based on PCR is simpler, quicker and less expensive. However, maybe we did not 
emphasize this comparison in the manuscript, therefore we explained it more in 
modified version. We also included information about how long the whole identification 
takes and how the costs are reduced. Hopefully, the Reviewer finds it appropriate to use 
inexpensive in the title of the paper.
2) The introduction and methods are well described. The conclusion should be
reformulated and improved.
We managed to improve the conclusions with the information discussed above, we hope 
that the Reviewer will find them better now.
All minor comments suggested by the Reviewer are addressed and can be followed in a 
modified manuscript in revision format. We decided to move Table 1 to supplementary 
materials and modify Table 2..
We believe that the revised manuscript is substantially improved, both in style and content, and hope 
that it is now acceptable for publication in the Pathogens Journal. If there are any additional questions, 
please do not hesitate to contact me. We look forward to hearing from you in due time.
Best regards,
Agnieszka Kajdanek

Reviewer 4 Report

Comments and Suggestions for Authors

I believe that paper of Agnieszka Kajdanek should be given a more modern approach. If we exclude the NGS, the rest of the techniques used are obsolete, or at least I think that introducing in the work figures with the development of a multiplex PCR I think it is at least anachronistic for a paper of 2024. I believe that the authors have done a great deal of work, but only in materials and methods. As from the written suggestions below, I believe that only the table concerning the results of the NGS and that of the oligos of the pcr, suitably modified as I suggested, should remain in the text. Finally, I wanted to point out that agarose gel is a detection technique that has been outdated for some years with capillary electrophoresis systems or similar.

Major points:

Table 1 should be made more essential in order to make it smaller: the full table may be included in the supplemental materials.

Table 2 modify the table so that it contains only the sequence info , the base pairs of the amplified and the region, possibly also indicating the corresponding genbank accession number.

Table 3 should be moved to supplemental materials

Figure 2 shall be deleted. Studies on the best PCR conditions can be cited in materials and methods!

In Figure 3 it is essential to mark the size in base pairs of each band of the molecular weight marker

Table 4 could be replaced by bars corresponding to each of a different type of bacterium corresponding to the positivity of each of the primer pairs used.

Finally, is it possible to visualize on the agarose gel any of the strains positive for more than one oligo pair?In addition, I would recommend sequencing all amplifiers obtained from individual PCRs, using the Sanger method: this would give a further confirmation of the results obtained by the NGS.

Author Response

Dear Reviewer,
Thank you for serving as the Reviewer to our manuscript pathogens-2949471 entitled: "A rapid and 
inexpensive PCR test for mastitis diagnosis based on NGS data". We wish to thank you for your 
supportive comments and constructive suggestions, and are pleased to inform you that in the enclosed 
revised manuscript we have addressed all the concerns which we agreed with. All changes appear in 
revision format in the revised manuscript. Attached please find a detailed point-by-point reply.
We are pleased that the Reviewer found our study in good quality and appreciate all his/her suggestions.
1) Table 1 should be made more essential to make it smaller: the full table may be included 
in the supplemental materials. 
We decided to move Table 1 to supplementary materials, as suggested.
2) Table 2 modify the table so that it contains only the sequence info, the base pairs of the 
amplified and the region, possibly also indicating the corresponding genbank accession 
number.
Table 2 was modified according to the Reviewer’s comment.
3) Table 3 should be moved to supplemental materials
We placed in supplemental materials.
4) Figure 2 shall be deleted. Studies on the best PCR conditions can be cited in materials 
and methods!
We followed these suggestions and appropriate changes were made.
5) In Figure 3 it is essential to mark the size in base pairs of each band of the molecular 
weight marker
It is now marked, we hope it looks better for the Reviewer.
6) Table 4 could be replaced by bars corresponding to each of a different type of bacterium 
corresponding to the positivity of each of the primer pairs used.
We decided to reorganize Table 4, marking just positive results and we hope that it is 
more clear now in the revised manuscript.
7) Finally, is it possible to visualize on the agarose gel any of the strains positive for more 
than one oligo pair? In addition, I would recommend sequencing all amplifiers obtained 
from individual PCRs, using the Sanger method: this would give further confirmation 
of the results obtained by the NGS
There was one such strain in our study: Staphylococcus sp. 043PP2022 positive for 2 
pairs of primers. It is a valuable suggestion from the Reviewer to sequence PCR 
products by the Sanger method to confirm our results and we surely going to consider 
it in our work.
All minor comments suggested by the Reviewer are addressed and can be followed in a modified 
manuscript in revision format.
We believe that the revised manuscript is substantially improved, both in style and content, and hope 
that it is now acceptable for publication in the Pathogens Journal. If there are any additional questions, 
please do not hesitate to contact me. We look forward to hearing from you in due time.
Best regards,
Agnieszka Kajdanek

Round 2

Reviewer 1 Report

Comments and Suggestions for Authors

No further comments.

Author Response

Dear Reviewer,

We would like to express our sincere gratitude for your valuable feedback. Your insights and suggestions are greatly appreciated.

Thank you very much.

Best regards,

Agnieszka Kajdanek

Reviewer 2 Report

Comments and Suggestions for Authors

In the revised version of the manuscript ID pathogens-2949471, the authors have tried to answer the reported concerns and modified the text accordingly, providing the motivations driving their experimental procedures. While this reviewer agrees with the aim of shortening the microbiological diagnosis process, the proposed study still lacks novelty, as different protocols have been described providing the same and even further advantages compared to the described PCR test (see the previous review). Since the authors often claim that such protocol could implement the pathogen detection in mastitis infections, some experiments testing DNA purified from milk samples are highly recommended to corroborate their thesis and to add significance to their study. Representative samples (e.g., milk spiked with the selected bacterial species) could be used, but the essential point is to test DNA coming from samples and not purified by bacterial isolates.

After performing this major revision, the paper can be considered for publication.

Author Response

Dear Reviewer,

We express our sincere gratitude for your comprehensive review and valuable insights. Your feedback is greatly appreciated and has provided us with a deeper understanding of the subject matter.

We concur with your suggestion to conduct PCR testing on milk samples and explore the use of primers in quantitative PCR (qPCR). This approach indeed holds substantial promise. However, we acknowledge that the process of testing and validating 100 samples would surpass our current response timeframe. To ensure rigorous validation of our test, collaboration with farmers and veterinarians is necessary for obtaining properly collected milk samples. Nevertheless, we perceive this as an exceptional opportunity for further enhancements to our test and potential content for a subsequent publication.

Our primary objective was to streamline the diagnostic process, maintain viable cultures, and eliminate biochemical tests that frequently yielded unsatisfactory results. This initiative stemmed from issues with incorrect species identification because the prevailing methodology did not meet our important criteria. Consequently, we did not directly isolate pathogens from the infected milk samples. The premise of this test was primarily accurate taxonomic identification. If we aim to determine the affiliation of a given bacterium to the species using this test, then isolating a pure culture is necessary. However, if we aim to check what is present in the milk, then it is not. We have exerted every effort to minimize the likelihood of obtaining false positive results for this PCR test. The analyzed amount of genomes and the conservatism of selected regions concerning a given species is so high that we believe the test should easily cope in field conditions on samples with ‘natural’ flora. However, some probability of false positive results always exists in such a large and complex system. Therefore, preliminary microbiological screening can only assist. However, a diagnostic test like this test necessitates further validation by the veterinary and scientific community, which is why we want to publish it and not commercialize it.

The innovation in our test lies in sample sequencing, leveraging the current taxonomy of bacterial strains. Unlike commercially available qPCR tests such as Mastitis 4BDF (DNA Diagnostics A/S) and PathoProof (ThermoFisher), which were developed over a decade ago based on 16S rRNA, our approach utilizes Next-Generation Sequencing (NGS) with enhanced accuracy and significantly reduces species misidentification.

Regarding the article you referenced, the immunochromatographic test served as an initial screening tool and could potentially be useful as an on-farm test. However, the authors still needed to confirm the results using PCR to ensure both sensitivity and specificity in pathogen detection.

As for the second article, they did employ one of the aforementioned kits, Mastitis 4BDF, for pathogen identification, which raises questions about the taxonomy. It’s worth noting that at the time, identification based on 16S rRNA was considered accurate. However, with the advent NGS technologies, we now can achieve significantly higher quality results. NGS provides a more comprehensive view of the microbial community, allowing for a more precise and detailed taxonomy.

Recognizing the importance of staying up-to-date, we aim to publish recent data promptly to ensure our findings remain relevant and do not become outdated.. Their input reinforces our confidence in the viability of our method.

Once again, we sincerely appreciate your valuable insights.

Yours faithfully,

Agnieszka Kajdanek

Reviewer 4 Report

Comments and Suggestions for Authors

I wanted to congratulate the authors as they followed my suggestions.

Author Response

(The authors gave the same response as above.)

Round 3

Reviewer 2 Report

Comments and Suggestions for Authors

In the newly revised version of the manuscript ID pathogens-2949471, the authors have tried to justify the lack of further, important experiments to provide more significance to their results. While this reviewer acknowledges the value of the NGS data, in the end the paper presents the validation of a multiplex PCR protocol applied to bacterial isolates. These results seem too preliminary to be published as article and need to be implemented with further data.